# *Akkermansia muciniphila* mediates negative effects of IFNγ on glucose metabolism

Renee L. Greer[1,*], Xiaoxi Dong[2,*], Ana Carolina F. Moraes[3], Ryszard A. Zielke[2], Gabriel R. Fernandes[4], Ekaterina Peremyslova[2], Stephany Vasquez-Perez[1], Alexi A. Schoenborn[5], Everton P. Gomes[6], Alexandre C. Pereira[6], Sandra R.G. Ferreira[3], Michael Yao[7], Ivan J. Fuss[7], Warren Strober[7], Aleksandra E. Sikora[2], Gregory A. Taylor[8], Ajay S. Gulati[5], Andrey Morgun[2,**] & Natalia Shulzhenko[1,**]

Cross-talk between the gut microbiota and the host immune system regulates host metabolism, and its dysregulation can cause metabolic disease. Here, we show that the gut microbe *Akkermansia muciniphila* can mediate negative effects of IFNγ on glucose tolerance. In IFNγ-deficient mice, *A. muciniphila* is significantly increased and restoration of IFNγ levels reduces *A. muciniphila* abundance. We further show that IFNγ-knockout mice whose microbiota does not contain *A. muciniphila* do not show improvement in glucose tolerance and adding back *A. muciniphila* promoted enhanced glucose tolerance. We go on to identify Irgm1 as an IFNγ-regulated gene in the mouse ileum that controls gut *A. muciniphila* levels. *A. muciniphila* is also linked to IFNγ-regulated gene expression in the intestine and glucose parameters in humans, suggesting that this trialogue between IFNγ, *A. muciniphila* and glucose tolerance might be an evolutionarily conserved mechanism regulating metabolic health in mice and humans.

[1] College of Veterinary Medicine, Oregon State University, 105 Dryden Hall, 450 SW 30th Street, Corvallis, Oregon 97331, USA. [2] College of Pharmacy, Oregon State University, 1601 SW Jefferson Way, Corvallis, Oregon 97331, USA. [3] Department of Epidemiology, School of Public Health, University of São Paulo, Av. Dr Arnaldo, 715, São Paulo, SP 01246-904, Brazil. [4] Oswaldo Cruz Foundation, René Rachou Research Center, Av. Augusto de Lima, 1715, Belo Horizonte, MG 30190-002, Brazil. [5] Division of Pediatric Gastroenterology, University of North Carolina at Chapel Hill, 260 MacNider Building, CB# 7220, Chapel Hill, North Carolina 27599, USA. [6] Laboratory of Genetics and Molecular Cardiology, Heart Institute (InCor), University of São Paulo Medical School, Av. Dr Eneas de Carvalho Aguiar, 44, São Paulo, SP 05403-000, Brazil. [7] Mucosal Immunity Section, Laboratory of Immune Defenses, National Institute of Allergy and Infectious Diseases, Bethesda, Maryland 20892, USA. [8] Geriatric Research, Education and Clinical Center, VA Medical Center, Departments of Medicine, Molecular Genetics and Microbiology and Immunology, Division of Geriatrics and Center for the Study of Aging and Human Development, Duke Box 3003, Duke University Medical Center, Durham, North Carolina 27710, USA. * These authors contributed equally to this work. ** These authors jointly supervised this work. Correspondence and requests for materials should be addressed to A.M. (email: anemorgun@hotmail.com) or to N.S. (email: natalia.shulzhenko@oregonstate.edu).

An important advance of the last couple of decades in biomedical science is the recognition that mammalian organisms do not function as a collection of functionally independent systems. Rather, there is extensive cooperation among systems that is essential for life, and its absence can result in dysfunction and disease. Numerous studies have revealed the involvement of the immune system in regulation of metabolism, and how the alteration of the immune system can contribute to metabolic abnormalities such as type 2 diabetes and metabolic syndrome[1–5]. These studies have primarily focused on immune cell effects on fat, liver and muscle, as besides the pancreas, these tissues are considered major metabolic organs responsible for glucose and lipid metabolism. One such example is the influence of IFNγ, which is a central cytokine of the immune system, on systemic glucose metabolism. Previous studies have shown that mice deficient in IFNγ have improved glucose tolerance[6–8]. Mechanistically, this phenomenon has been attributed to reduced hepatic glucose production[6] and increased insulin sensitivity, possibly related to reduced adipose inflammation in case of obese animals[8].

More recently, the gut has emerged as an important player in systemic metabolism. Besides producing several hormones, the gut harbours thousands of microbes (the gut microbiota) which themselves function as a metabolically active organ[9,10]. Therefore, by modulating the composition and dynamics of the gut microbiota, the immune system may ultimately exert a major impact on the metabolism of the organism. A few recent studies have demonstrated physiologically important trialogues among the immune system, gut microbiota and metabolism[11–14]. However, despite the emerging evidence of importance of such trialogues, much research continues to focus on two-component dialogues, thus failing to appreciate the complete picture of communication between multiple systems.

In the current study, we addressed whether the established dialogue between IFNγ and glucose metabolism involves a third player—the gut microbiota. By using systems biology approaches and analysing transkingdom interactions we found that, indeed, the effect of IFNγ on glucose tolerance is mediated by one of the members of mouse gut microbiota, *A. muciniphila*. Further, we have identified immunity-related GTPase family, M (Irgm1) as an IFNγ-regulated host gene responsible for control of *A. muciniphila* levels in the gut. In addition, the investigation of human subjects revealed that *A. muciniphila* may play similar roles in mouse and human physiology.

## Results

**IFNγ-regulated bacterial modulators of glucose metabolism.** Similar to previous reports[6–8], we observed that glucose tolerance is significantly improved in IFNγKO mice (Fig. 1a). To start addressing our hypothesis that gut microbiota is a mediator of effect of IFNγ on glucose metabolism, we first treated wild-type (WT) and IFNγKO mice with a cocktail of antibiotics that has been successfully employed in previous studies to eliminate the majority of gut bacteria to test their role in host physiology[15–17]. Overall, glucose metabolism was improved following antibiotic treatment in both genotypes (Fig. 1a), which is consistent with previous findings that, as a whole, microbiota worsen glucose metabolism[18–21]. Importantly for this study, treatment with antibiotics abolished differences between the two genotypes, supporting our hypothesis that microbiota mediate the effect of IFNγ on glucose metabolism (Fig. 1a). Body weight and food intake alone could not consistently explain differences in glucose tolerance (Supplementary Fig. 1).

We next sought to determine microbe(s) mediating effect of IFNγ on glucose metabolism. Such microbes would need to fulfil two criteria: (1) to be regulated by IFNγ and (2) to regulate glucose metabolism. Thus, in the exploratory phase, we first assessed which microbes were differentially abundant under IFNγ perturbation. Next, in a separate set of analyses using correlations with metabolic measurements, we identified which of the IFNγ-regulated bacteria could be potential regulators of glucose metabolism (Fig. 1b). To identify such microbes and to minimize confounding effects, we used two independent methods to perturb IFNγ levels—genetic disruption of IFNγ and blockade with anti-IFNγ antibody (Fig. 1b). When microbial abundances between IFNγKO and corresponding wild-type mice were compared by sequencing of the bacterial 16S ribosomal RNA (rRNA) gene, 555 differentially abundant operational taxonomic units (OTUs) were identified, corresponding to 33 different genera (Supplementary Fig. 2A, Supplementary Data 1). Next, to narrow and validate our findings, we used a second method to perturb IFNγ levels. We took advantage of the fact that germfree mice have very low levels of systemic and intestinal IFNγ and that microbiota induce expression of this cytokine in the gut[22] (Supplementary Fig. 2E). We colonized wild-type germfree mice with microbiota from IFNγKO mice and blocked the rising levels of IFNγ with an anti-IFNγ antibody to maintain low levels during colonization while a control group was treated with rat IgG (Supplementary Fig. 2E). We reasoned that taxa that have similar differential abundance in both experiments (genetic knockout and antibody blockade) are more likely to be regulated by IFNγ. As expected, we observed a significant increase in IFNγ levels 7 days after colonization that was prevented by anti-IFNγ antibody injection (Supplementary Fig. 2E). Sequencing of the 16S rRNA in caecum revealed that 248 OTUs were differentially abundant (Supplementary Fig. 2D, Supplementary Data 2), of which 69 OTUs were concordant with the IFNγKO versus WT results (Fig. 1c, Supplementary Data 3).

Once we identified IFNγ-regulated bacteria, we searched for those that would be predicted to mediate the effect of the cytokine on glucose metabolism. To achieve this, we analysed correlations between the abundance of IFNγ-regulated microbes and glucose metabolism parameters such as fasted glucose levels and area under curve of glucose tolerance test (AUC-GTT). This analysis was performed in IFNγKO mice so that direct effects of IFNγ could not bias the correlation. With this approach[23], microbial candidates that mediate the effect of IFNγ on glucose metabolism should present a positive correlation with glucose levels and AUC-GTT if they are enriched in the presence of IFNγ, and negative correlation if they are depleted by IFNγ (see experimental outline in Supplementary Fig. 3). Through this analysis we identified four different OTUs, all corresponding to *A. muciniphila*, as top candidate improvers of glucose metabolism (Fig. 1d). Increased abundance of *A. muciniphila* is detected in both the ileum and stool of IFNγKO mice, and levels in the stool are representative of those in the ileum (Fig. 1e, Supplementary Fig. 2B,C) and correlate to fasting glucose and AUC-GTT (Fig. 1f,g). In addition, one OTU corresponding to Bacteroidetes S24-7, which could not be assigned to a specific taxon, was identified as a top candidate for worsening of glucose tolerance metrics (Fig. 1d).

**A. muciniphila mediates effect of IFNγ on glucose tolerance.** *A. muciniphila* is a well-known, cultivable species present in both the mouse and human microbiota[24]. Interestingly, *A. muciniphila* has previously been linked to metabolism—it is reduced in obese mice and patients, and restoration of its levels improves glucose metabolism in mouse models of metabolic disease[25–27]. Our analysis, thus far, predicted *A. muciniphila* as a key candidate for improvement of glucose tolerance in lean IFNγKO mice. To validate this relationship, we performed a series of confirmatory

loss- and gain-of-function experiments (Fig. 2a). First, we restored IFNγ levels in KO mice by administering exogenous recombinant IFNγ. All IFNγKO mice showed identical initial glucose tolerance (Fig. 2b) at the start of the study. Following

2 weeks of injections, serum IFNγ levels were elevated compared with PBS control, but did not reach wild-type levels; therefore it is unlikely that activation of IFNγ pathways was induced above what would be expected for wild-type mice (Fig. 2f). Mice that

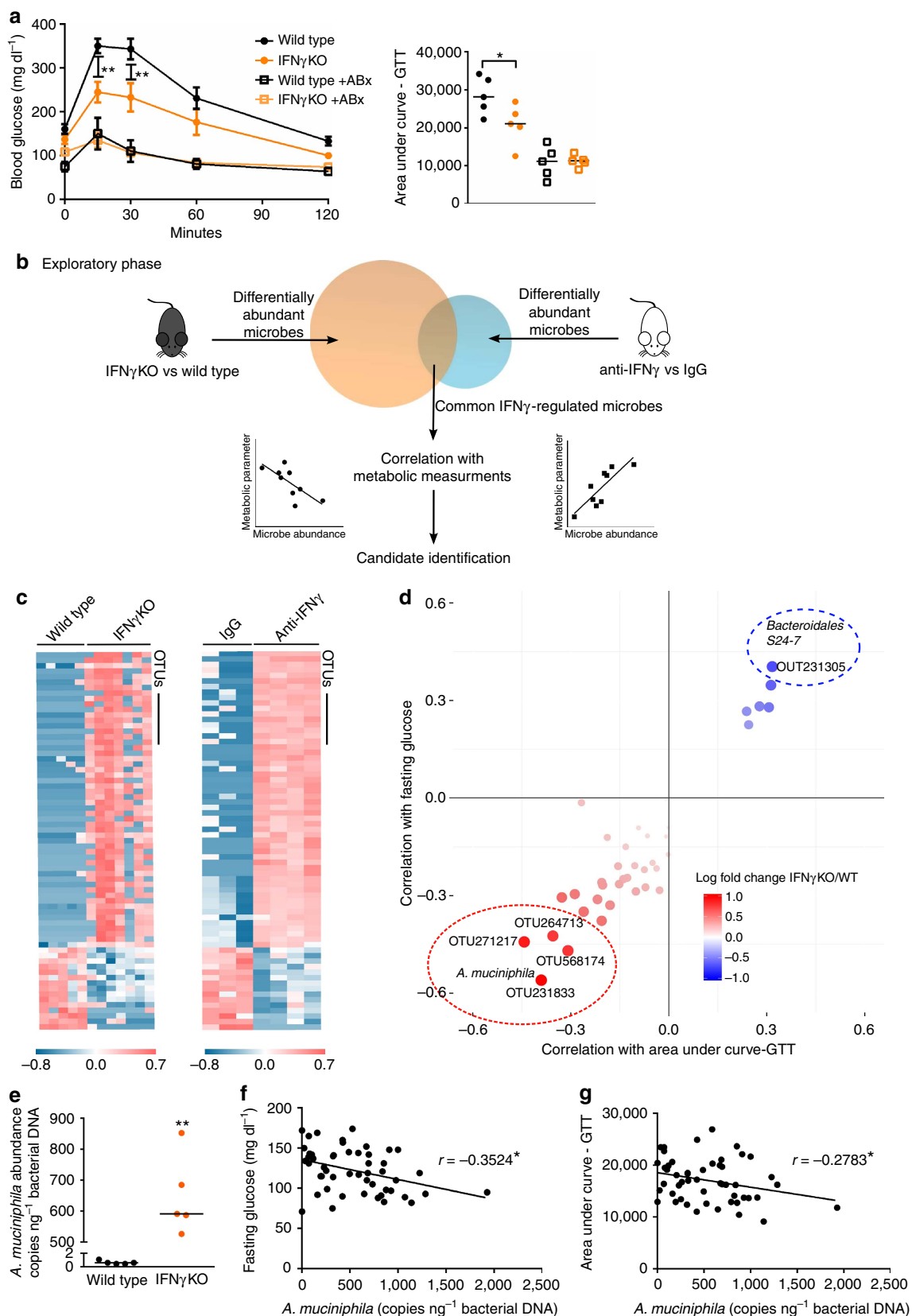

received IFNγ showed significantly worse glucose tolerance than PBS controls (Fig. 2c), coincident with a decrease in abundance of *A. muciniphila* levels (Fig. 2d). These data demonstrate the ability of IFNγ to regulate *A. muciniphila* as well as to regulate glucose tolerance, but do not rule out the possibility that these two effects are independent.

Next, to directly test if IFNγ acts through *A. muciniphila* as our predictive analysis suggests, we bred IFNγKO mice with *A. muciniphila*-negative wild-type mice to generate *A. muciniphila*-negative IFNγ heterozygotes, which were then interbred to ultimately obtain *A. muciniphila*-negative IFNγKO mice (IFNγKO/*Akk_neg*) that was possible due to lack of exposure from heterozygous parents (Fig. 2a middle panel; Supplementary Fig. 4D). After three generations of breeding, we achieved close to non-detectable levels (<1 copy per ng bacterial DNA) of *A. muciniphila* in the stool of IFNγKO mice (Supplementary Fig. 4D). There was no difference in glucose tolerance between wild-type and IFNγKO/*Akk_neg* mice (Fig. 3b), demonstrating that by removal of *A. muciniphila* from the system we could abrogate the effect of IFNγ on glucose levels. However, we reasoned that the breeding strategy might have altered the abundance of other taxa in gut microbiota in addition to *A. muciniphila*. Therefore we performed 16S rRNA gene profiling of the IFNγKO/*Akk_neg* microbiota compared with natively *A. muciniphila* positive mice from Jackson Labs. We identified only three taxa other than *A. muciniphila* to be different following this breeding strategy (Supplementary Table 1). Therefore, to test whether *A. muciniphila*, and not some other altered taxa, was causative of metabolic improvement in IFNγKO mice, we reconstituted a subset of IFNγKO/*Akk_neg* mice with *A. muciniphila* (IFNγKO/*Akk_pos*) (Fig. 3e). Seven days after colonization we observed better systemic glucose tolerance in IFNγKO/*Akk_pos* mice, while IFNγKO that did not receive *A. muciniphila* continued presenting glucose tolerance similar to wild-type mice (Fig. 3d, Supplementary Fig. 4), thus confirming that *A. muciniphila* is sufficient to mediate the effects of IFNγ on systemic glucose metabolism. Finally, we restored IFNγ levels in these IFNγKO/*Akk_neg* and IFNγKO/*Akk_pos* mice through injection of recombinant IFNγ. Only mice carrying *A. muciniphila* responded to treatment by worsening of glucose tolerance (compare Fig. 3f to Fig. 3h, Supplementary Fig. 4), thus demonstrating that IFNγ acts by controlling *A. muciniphila* to worsen glucose tolerance in IFNγKO mice. As previous studies primarily linked *A. muciniphila* to glucose metabolism in obese mice[26,27], we also tested its ability to improve glucose metabolism in lean wild-type mice. Indeed, administration of *A. muciniphila* enhanced glucose tolerance in lean wild-type mice (Supplementary Fig. 5).

It is possible that the administration of *A. muciniphila* may have altered the abundance of other microbes that could, in turn, alter glucose tolerance. Therefore, we performed analysis to identify microbes that are potentially regulated by *A. muciniphila* and have evidence to be related to glucose metabolism. We identified three microbial genera that showed different abundance after *A. muciniphila* colonization and correlation to glucose tolerance, including *Akkermansia*, (False discovery rate (FDR) < 0.1; Supplementary Table 2). *A. muciniphila* presented the strongest and most significant correlation. However, these other microbes might be interesting areas of further study. It is also possible that IFNγ injection altered microbes in addition to *A. muciniphila*. Therefore we performed a similar analysis as above, comparing taxa abundance before and after injection within IFNγKO/*Akk_pos* mice. Although some minor trends of alteration of microbe abundance were observed, no genera except *A. muciniphila* were significantly altered by rIFNγ injection at FDR < 0.1. Therefore, although we cannot rule out a role for other microbes in mediating glucose tolerance upon administration of *A. muciniphila* and following injection of rIFNγ, our analysis did not provide any plausible candidate that may play a role in glucose tolerance responses.

**Irgm1 is a mediator of the effect of IFNγ on *A. muciniphila*.** Now that we have established *A. muciniphila* as a main contributor to improved glucose tolerance in IFNγKO mice, the question remained how IFNγ controls *A. muciniphila* levels. IFNγ has a central role in orchestrating response to multiple gut microbes by driving different effector mechanisms[28]. To identify genes mediating effect of IFNγ on *A. muciniphila*, we employed a comprehensive approach by measuring global gene expression. As a first step of our analysis we searched for mouse genes whose expression is regulated by IFNγ in the ileum, but not dependent on the presence of *A. muciniphila* in the gut microbiota (that is, genes located downstream of IFNγ and upstream of *A. muciniphila*). To detect these genes we compared ileal gene expression between wild-type, IFNγKO/*Akk_neg* and IFNγKO/*Akk_pos* mice. These analyses revealed 229 differentially expressed genes (FDR < 0.1) between wild-type and IFNγKO mice regardless of *A. muciniphila* status (Fig. 4a).

Network analysis has been an efficient tool in the identification of physiological processes and finding causal genes in host–microbe interactions[15], as well as in cancer[29,30]. Therefore, we reconstructed a gene network of the IFNγ-dependent mouse ileum transcriptome which was comprised of 165 out 229 differentially expressed genes (Fig. 4b). As it could be expected, most of these genes had lower expression in IFNγKO mice compared with controls. The interrogation of the network revealed overrepresentation of Gene Ontologies for immune responses including MHC (major histocompatibility complex) Class I antigen presentation, T cell activation and interferon-inducible GTPase (Supplementary Data 4). Furthermore, among top hub genes (high connectivity degree) that usually consist of upstream regulators were *Stat1*, *Igtp*, *Tap1* and other genes representing the aforementioned immune pathways (Supplementary Data 4).

Often further investigation is focused on hub genes because of their potential probability to be master regulators of

**Figure 1 | Identification of *A. muciniphila* as a predicted IFNγ-dependent regulator of glucose tolerance.** (**a**) Intraperitoneal glucose tolerance test (IP-GTT) and area under the curve quantification in conventional IFNγKO and wild-type control mice before (closed circles) and after (open squares) 2 weeks of antibiotic cocktail treatment (n = 5 per group). Glucose tolerance curves shown as mean ± s.e.m., median line is displayed on dot plots. (**b**) Experimental outline describing the exploratory phase for prediction of IFNγ-regulated microbes that are modulators of glucose metabolism. (**c**) Heat maps of common differentially abundant microbes in IFNγKO versus wild-type stool and anti-IFNγ versus IgG caecal content. Differentially abundant microbes are selected based on *t*-test FDR < 0.1. (**d**) Correlation of differentially abundant microbes to area under curve of glucose tolerance (AUC-GTT) test and fasting glucose. Colour intensity indicates direction of change of microbe in IFNγKO versus wild type (red = more abundant in IFNγKO). Size of each point indicates Spearman correlation *P* value with larger spots representing higher significance. Dashed circles indicate *P* value cutoff of 0.05. All four points within the red circle are unique OTUs, all representing *A. muciniphila*. (**e**) Quantification of *A. muciniphila* copy number by qPCR, represented as copies *A. muciniphila* genome per ng total 16S DNA. (n = 5 per group, one representative experiment out of 3). (**f,g**) Spearman correlation of *A. muciniphila* copies per ng bacterial DNA with fasting glucose (**f**) and area under the curve of glucose tolerance test (**g**) in IFNγKO mice (n = 50). *P < 0.05, **P < 0.01, ***P < 0.001 by one-tailed Mann–Whitney test except where indicated otherwise.

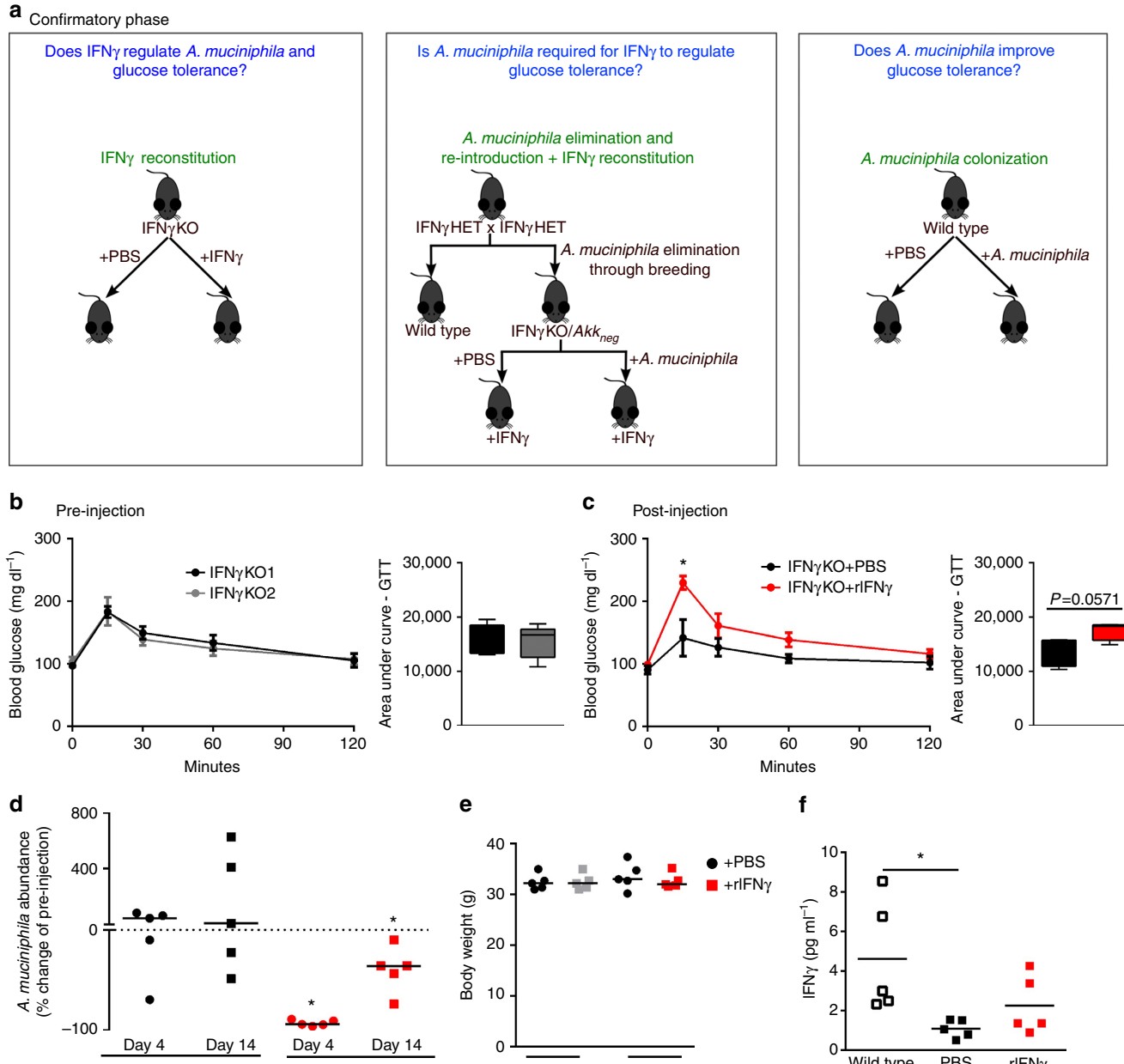

**Figure 2 | IFNγ reconstitution validates IFNγ as a regulator of *A. muciniphila* and glucose tolerance. (a)** Experimental outline describing the confirmatory phase where the identified candidate from Fig. 1b exploratory phase, *A. muciniphila*, is directly tested by three independent approaches. Readouts of all experiments are quantification of *A. muciniphila* abundance and assessment of glucose tolerance. **(b,c)** IP-GTT and area under the curve of IFNγKO mice before **(b)** and following 2 weeks of rIFNγ or PBS administration **(c)**. **(d)** *A. muciniphila* was quantified by qPCR. Shown is percent change of *A. muciniphila* abundance in stool from initial pre-injection levels after the 2-week injection period. **(e)** Body weight of all groups of mice pre- and post-injection. **(f)** Serum IFNγ levels at the post-injection time point. Glucose tolerance curves shown as mean ± s.e.m., box plots represent median with 25th and 75th percentile borders and error bars represent min–max. Median line is displayed on dot plots. For all glucose tolerance tests and qPCR results shown, $n = 5$ per group. *$P < 0.05$ by one-tailed Mann–Whitney test.

processes[29,31]. In this study, however, we were specifically interested in IFNγ-dependent genes positioned at the interface of the host gene regulatory network and *A. muciniphila*. To infer these genes, we again used causal inference analysis similar to that which was previously described for *A. muciniphila* discovery. In this analysis, we derived a ranking calculation that considered differential gene expression, correlation of each gene to *A. muciniphila* levels and peripheral-ness of a gene in the network (see 'Methods' section for complete details). This analysis revealed a few potential inhibitors of *A. muciniphila*,

with *Irgm1* being the top ranked candidate by this index (Fig. 4c,d). Next we tested the prediction of *Irgm1* being an inhibitor of *A. muciniphila* by comparing abundance of this microbe between Irgm1 knockout mice (Irgm1KO) and control mice in two different mouse facilities. Notably, despite a large difference in overall *A. muciniphila* abundance between the two sites, Irgm1KO mice had increased abundance of this microbe compared with their corresponding wild-type controls (Fig. 4e). To validate that this increase was due to the absence of Irgm1 and not a feedback loop altering IFNγ

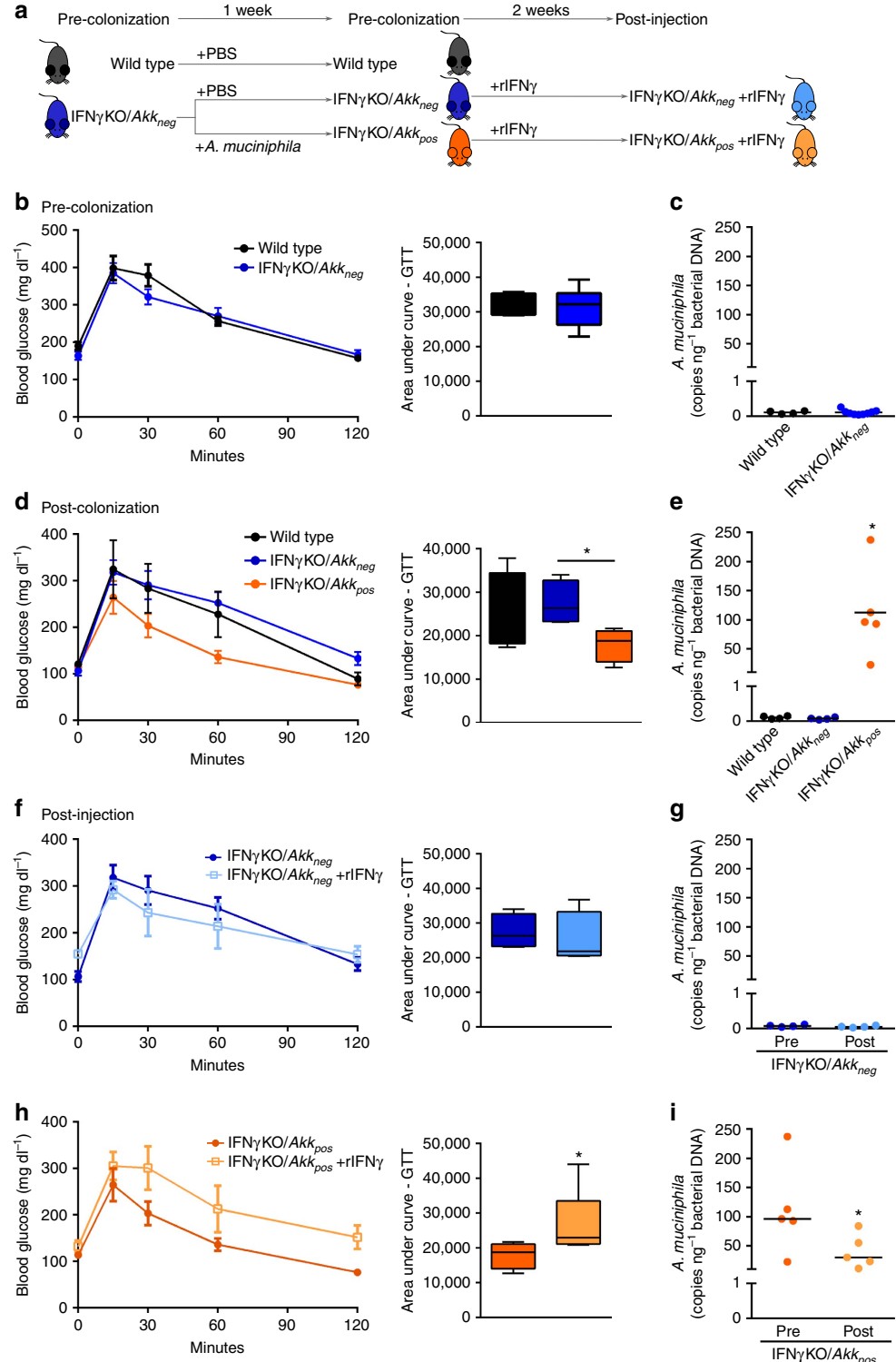

**Figure 3 | A. muciniphila is required for IFNγ regulation of glucose tolerance.** (a) Experimental scheme: *A. muciniphila*-negative wild-type and IFNγKO mice were colonized with either PBS or *A. muciniphila* and subsequently injected with recombinant IFNγ (rIFNγ). (b,d) Pre-colonization (b) and post-colonization (d) IP-GTT. (c,e) Pre-colonization (c) and post-colonization (e) *A. muciniphila* levels by qPCR expressed as copies of *A. muciniphila* per ng total bacterial DNA. (f,h) IP-GTT in IFNγKO/*Akk_neg* (f) and IFNγKO/*Akk_pos* (h) before and after 2 weeks of injection with rIFNγ. Darker shades represent before injection and represent the same test shown in d for each respective group; lighter shades represent after injection. (g,i) *A. muciniphila* levels by qPCR expressed as copies of *A. muciniphila* per ng total bacterial DNA in IFNγKO/*Akk_neg* (g) and IFNγKO/*Akk_pos* (i) before and after 2 weeks of injection with rIFNγ. Darker shades represent before injection, lighter shades represent after injection. Glucose tolerance curves shown as mean ± s.e.m., box plots represent median with 25th and 75th percentile borders and error bars represent min–max. Median line is displayed on dot plots. At pre-colonization time point $n = 4$ for wild type and 9 for IFNγKO/*Akk_neg*. At post-colonization and post-injection time points, $n = 4$ for wild type and IFNγKO/*Akk_neg* and 5 for IFNγKO/*Akk_pos*. *$P < 0.05$ by one-tailed Mann–Whitney test.

signalling overall, we examined global gene expression in the ileum of these mice. Overall, very few genes from our previously identified IFNγ-dependent network were significantly altered (Supplementary Data 5). Notably, IFNγ itself was not changed, nor were any of our top candidate *A. muciniphila* regulators from our previous network analysis (Fig. 4f). Thus, these results corroborate our computational prediction

that Irgm1 is a significant factor in regulation of *A. muciniphila* by IFNγ.

**A. muciniphila relates to glucose and IFNγ in humans.** *A. muciniphila* is also a frequent resident of the human gut microbiome[24]. Therefore, we took advantage of a cohort of subjects enrolled by Brazilian Advento Study Group to see if the

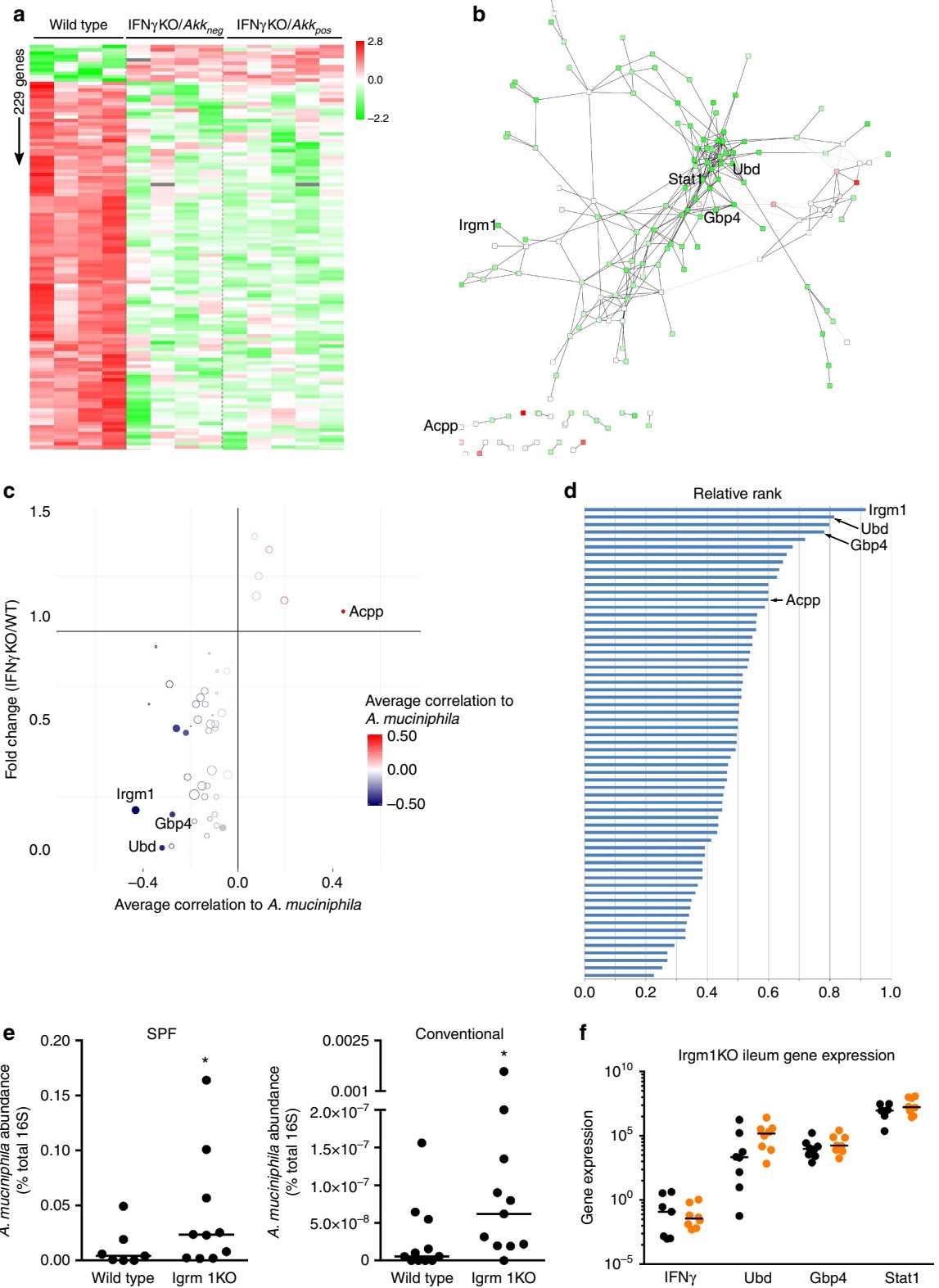

relation between *A. muciniphila* and metabolism we observed in mice can be also found in human population. Investigations have recently related levels of *A. muciniphila* with diabetes and/or obesity[25,32–34], however, several other metagenomic studies did not report an association between this bacterium and metabolic abnormalities in humans[35,36]. Considering that multiple gut microbes besides *A. muciniphila* may influence glucose metabolism, we speculated that in cases where this microbe is at low levels, it is less likely to contribute considerably to the phenotype because other more abundant microbes would be stronger players. To define biologically significant levels, we referred back to our IFNγKO mice that presented negative correlation between *A. muciniphila* and fasting glucose levels (Fig. 1f,g). The abundance of *A. muciniphila* was >1% in the majority of those mice (Supplementary Fig. 2). Therefore, from the total of 295 human subjects we selected those with abundance of *A. muciniphila* ≥1% ($N = 94$). We found that *A. muciniphila* had a weak but significant negative correlation with glucose and glycated haemoglobin (HbA1c) (Spearman $r = -0.3167$ $P < 0.001$ and $r = -0.3033$ $P < = 0.01$, respectively) (Fig. 5a,b). We then used American Diabetes Association guidelines[37,38] for classification of these subjects into three groups based on glycaemia status by fasting plasma glucose, 2 h plasma glucose (2-PG) and HbA1c and assessed *A. muciniphila* abundance in these groups. Individuals with normal glucose metabolism showed significantly higher *A. muciniphila* abundance compared with type 2 diabetics, with pre-diabetics showing an intermediate abundance of *A. muciniphila* (Fig. 5c). In the group with diabetes, some patients were on treatment with metformin, which had been previously associated with increased *A. muciniphila* in mice[27,39]. However, we did not detect differences for *A. muciniphila* abundance, fasting glucose or HbA1c between subjects treated or not treated with metformin (Supplementary Fig. 6). While these results require a confirmation in independent human cohorts, they support the idea that *A. muciniphila* may play a similar role in mice and humans in regulation of glucose metabolism.

Data regarding intestinal expression of IFNγ was not available in human subjects that were evaluated for faecal microbiome and glucose metabolism. Therefore, we turned to another group of human subjects in whom we had measured global gene expression and *A. muciniphila* levels in duodenal biopsies. This group of subjects consisted of three subgroups including healthy volunteers, and two different groups of patients with common variable immunodeficiency. Analysis showed a trend to a negative correlation (Pearson $r \approx -0.3$, $P = 0.127$) between IFNγ gene expression and *A. muciniphila* levels (Fig. 5d, top gene). Therefore, we decided to analyse the human gene signature corresponding to mouse homologues we have defined as stimulated by IFNγ in the murine intestine (Fig. 4b). Out of

about 220 mouse genes, we found 162 human homologues with 141 of them being detectable in duodenal biopsies.

Analysing the correlation between expression of these genes and *A. muciniphila* levels, we found that approximately half of the gene signature (69 genes) had the same signs of correlations in all three analysed groups of subjects. These were all negative correlations with no gene presenting a consistent (through all three groups) positive correlation (Fig. 5d, test for one proportion $P < 0.0001$, Supplementary Data 6). Thus, despite small sample sizes in each individual group, the combined analysis showed consistent negative correlation for several IFNγ-dependent genes supporting the hypothesis that IFNγ may contribute to control of *A. muciniphila* levels not only in mice, but also in humans.

## Discussion

Our study has uncovered a missing link between IFNγ and glucose metabolism by demonstrating that a gut commensal, *A. muciniphila*, is a key microbe responsible for improved glucose tolerance observed in IFNγKO mice (Fig. 5e). Notably, two primary players that have been revealed to mediate the effect of IFNγ (Irgm1 and *A. muciniphila*) could not have been easily predicted based solely on the existing knowledge in the field. Rather, the generation of testable hypotheses in both cases was mainly a result of causal inference involving trankingdom network analysis that we have recently developed (reviewed in ref. 40). This approach has been previously successful in finding microbes and microbial genes that affect host phenotype[15]. This is the first time, however, when such strategy aided in prediction of host gene controlling a specific member of gut microbiota.

It is well established that IFNγ is important for control of multiple, primarily intracellular, pathogens. The effect of this cytokine on gut microbiota, however, has not been explored. Using two methods (genetic deletion and blocking antibody) we revealed that multiple OTUs from commensal microbiota were affected by IFNγ (Fig. 1, Supplementary Data 1–2). Following identification of IFNγ-regulated microbes, causal inference analysis allowed us to discern candidates relevant to the phenotype of interest (that is, glucose levels). We then validated the prediction that *A. muciniphila* is a mediator of effect of IFNγ on glucose metabolism by colonizations of different hosts with *A. muciniphila* and reconstitution of IFNγKO mice with recombinant IFNγ.

Altogether, the colonization of IFNγKO and wild-type mice with *A. muciniphila* shows that that this bacterium can improve glucose metabolism (fasting glucose and glucose tolerance) in different hosts. We cannot, however, make a definitive conclusion which other microbes might be required for its effect on glucose metabolism.

**Figure 4 | IFNγ regulates *A. muciniphila* abundance through Irgm1.** (**a**) Heat map of transcript abundance of IFNγ-dependent genes. Genes that show differential abundance between wild type and IFNγKO (*t*-test FDR < 0.1), but no difference between IFNγKO/*Akkneg* and IFNγKO/*Akkpos* (*t*-test FDR < 0.1) are shown. (**b**) Network reconstruction of IFNγ-dependent genes shown in **a**. Colours indicate fold change of expression as indicated in **a**. A file containing complete information for this network is available for download upon request. (**c**) Correlation of IFNγ-dependent genes with *A. muciniphila* levels. Pearson correlation between ileum *A. muciniphila* abundance and gene expression were calculated in three groups separately and the average correlation coefficient was shown. Colour intensity of each point indicates strength of correlation to *A. muciniphila* levels. Size of each point indicates average shortest path length, with larger points representing longer paths. (**d**) Ranking of IFNγ-dependent genes as potential regulators of *A. muciniphila*. Ranking takes into account strength of correlation with *A. muciniphila* and average shortest path length, with longer path lengths (that is, more peripheral to the network) resulting in higher ranking scores. See 'Methods' section for a more detailed description of calculation of this rank score. (**e**) *A. muciniphila* abundance in Irgm1KO mice housed in specific pathogen free conditions (*n* = 7 wild type, 10 Irgm1KO) and conventional conditions (*n* = 11 per genotype) by qPCR, represented as per cent *A. muciniphila* of total 16S rRNA DNA. (**f**) Gene expression of top IFNγ-dependent candidate genes from **d** determined by RNA-seq in the Irgm1KO ileum under specific pathogen free conditions; *n* = 7 wild type (black symbols), 10 Irgm1KO (orange symbols). *Acpp*, acid phosphatase, prostate; *Gbp4*, guanylate binding protein 4; *Irgm1*, immunity-related GTPase family, M; *Stat1*, signal transducer and activator of transcription 1; SPF, Specific pathogen free; *Ubd*, Ubiquitin D. Median line displayed on dot plots. *P < 0.05 by one-tailed Mann–Whitney test.

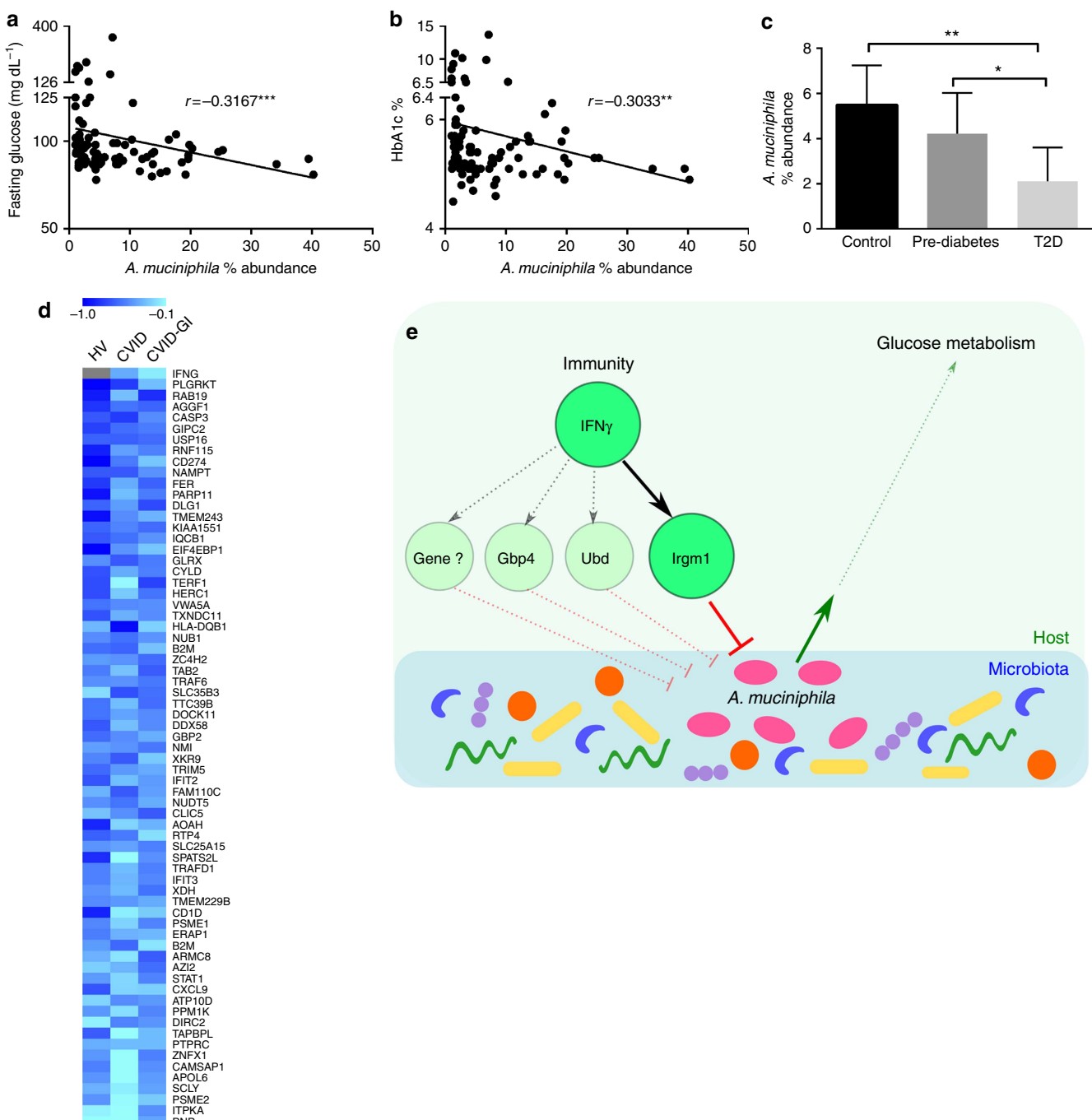

**Figure 5 | A. muciniphila correlates to glucose measures in human subjects and is reduced in diabetic patients.** (**a,b**) Spearman correlation of *A. muciniphila* percent abundance with fasting glucose (**a**) and HbA1c (**b**) in participants in the Advento Study ($n = 94$) (**c**) *A. muciniphila* percent abundance in normal ($n = 58$), pre-diabetic ($n = 31$) and type 2 diabetic subjects ($n = 11$). Bar plot represents mean and 95% confidence interval. Significance assessed by one-tailed Mann–Whitney test. (**d**) Heat map of Pearson correlation coefficients between each individual IFNγ-dependent gene and abundance of *A. muciniphila* of duodenal biopsies in three groups of samples. Individual $P$ value $< 0.2$, combined FDR $< 0.1$ for 59 out of 69 genes (Supplementary Data 6); genes ranked by strength of correlation according to Fisher's combined probability test. Grey colour indicates that a gene was below the level of detection. $*P < 0.05$, $**P < 0.01$, $***P < 0.001$. (**e**) Graphical model for regulation of glucose metabolism by IFNγ through the microbiota. IFNγ regulates expression of genes such as Irgm1 and Gbp4, which in turn, contribute to regulation of *A. muciniphila* levels in the gut. Differences in *A. muciniphila* abundance ultimately result in differences in systemic glucose tolerance in the host, with higher abundance of *A. muciniphila* inducing improvement of tolerance. CVID, Common Variable Immunodeficiency; CVID-GI, CVID with gastrointestinal symptoms; HV, healthy volunteer.

Although, in the current study we did not investigate which type of immune cells are of the source(s) IFNγ, intraepithelial T lymphocytes are the most plausible candidates. Besides their ability to produce IFNγ, intraepithelial T lymphocytes are the strongest responders among cells of adaptive immune system to changes in the microbiota[15]. This agrees with a recent study demonstrating that *A. muciniphila* levels are higher in mice deficient of T and B lymphocytes (Rag1KO) than in wild-type mice[41].

We also inferred and validated a molecule downstream of IFNγ, Irgm1, as a regulator of *A. muciniphila*. Although Irgm1 has been previously implicated in the control of intracellular pathogens[42,43], Irgm1KO mice also have Paneth cell abnormalities[44]. Because the secretion of antimicrobial proteins from Paneth cells is induced by IFNγ (ref. 45), we can speculate that Irgm1 may be a part of this signalling cascade. Ultimately, impaired production of antimicrobial peptides in the gut of IFNγKO mice may be a potential mechanism leading to outgrowth of *A. muciniphila*.

The second most favourable candidate among those identified as host genes-regulators of *A. muciniphila* is ubiquitin D (*Ubd* or *FAT10*; Fig. 4c,d), Interestingly, disruption of Ubd in mice has been shown to improve glucose tolerance along with other metabolic parameters but the impact on gut microbiota has not been examined[46]. Thus, while we have shown that Irgm1 is a mediator of the effect of IFNγ on *A. muciniphila,* it is plausible that other IFNγ-dependent mechanisms may also contribute to this phenomenon.

Our results from human subjects demonstrate that *A. muciniphila* regulation of metabolism may be an evolutionarily conserved mechanism between mice and humans. Relevance of the trialogue (IFNγ→*A. muciniphila*→glucose metabolism) to human health is further supported by evidence of increased levels of IFNγ producing cells in diabetes[47,48] and decrease abundance of *A. muciniphila*[25,32–34] in obese and diabetic patients. Interestingly, *A. muciniphila* levels have been recently demonstrated to negatively correlate with several inflammation markers associated with metabolic disease in mice[49]. Overall, these results suggest that loss of this bacterium can be due to local immune activation in the gut during disease and that this loss has implications for systemic metabolism. This topic warrants further investigation that should involve comprehensive evaluation of patients' immune status including in intestinal tissues.

Our findings may also explain response of mice to metformin, the most widely used drug for type 2 diabetes, that was also shown to block IFNγ production[50] and to increase levels of *A. muciniphila* in mice[27]. However, this particular mechanism might be different in mice and humans because neither our data (Supplementary Fig. 6) nor other more comprehensive human studies[51] found association between *A. muciniphila* and treatment with metformin.

Finally our study revealed a new homeostatic regulatory process in mammalian organisms, where a member of different kingdom, *A. muciniphila,* constitutes an integral part of the interaction between the supposedly functionally distinct and distant systems of immunity and glucose metabolism. Furthermore, our results and other published work in mouse models and human subjects suggest that this transkingdom interaction may be common in mammals[25–27]. Over the years, biologists have drawn boundaries between systems and kingdoms. Our results highlight the fact that these boundaries must be crossed to fully understand the complexity of living organisms.

## Methods

**Mice.** IFNγ knockout (IFNγKO on C57BL/6J background) and C57BL/6J controls were initially purchased from The Jackson Laboratory (Bar Harbor, Maine). Mice were housed at the Laboratory Animal Resource Center at Oregon State University under standard 12-h light cycle with free access to food (5001, Research Diets) and water. For all colonization studies, mice were maintained with autoclaved supplies, food (5010, Research Diets) and water. Adult mice of 8–10 weeks were used for all studies. Male mice were used for metabolic experiments, while males and females were used for microbiota sequencing and gene expression experiments. For experiments with IFNγKO/*Akk*$_{neg}$ mice, C57BL/6J and IFNγKO originally purchased from Jackson Labs were bred to generate heterozygous IFNγHET mice. There heterozygous mice were then interbred for two generations to IFNγKO/*Akk*$_{neg}$ mice. Experimental procedures were carried out in accordance with

protocols approved by the Oregon State University Institutional Animal Care and Usage Committee. Antibiotics were administered in drinking water for 2 weeks in the following concentrations: ampicillin ($1\,g\,l^{-1}$), vancomycin ($0.5\,g\,l^{-1}$), neomycin trisulfate ($1\,g\,l^{-1}$) and metronidazole ($1\,g\,l^{-1}$). Irgm1KO mice generated and maintained at the Durham VA and Duke University Medical Centers in conventional and specific pathogen free colonies. These mice have been described previously[42,43] and were backcrossed to C57Bl/6NCr1 mice for nine generations. Use of the Irgm1 mice was approved by the IACUC of the Durham VA and Duke University Medical Centers.

**Bacterial culture.** *A. muciniphila* ATCC BAA-835 was streaked out from $-80\,°C$ on BD Brain Heart Infusion (BHI) agar supplemented with 0.4% mucin (Sigma) and incubated under anaerobic conditions using the GasPack 100 system (BD Biosciences) at $37\,°C$ for 36 h. Bacterial colonies were swabbed from the plates, suspended in liquid BHI medium and 100 μl of the solution was plated on BHI agar containing 0.4% mucin. After 36 h of incubation at $37\,°C$ in anaerobic jar, bacteria were swabbed from plates, suspended in 10 ml of BHI containing 15% glycerol, aliquoted and stored at $-80\,°C$. To determine the colony forming units, one aliquot was thawed, serially diluted and plated on BHI agar, and bacterial colonies were enumerated after 36 h.

**Anti-IFNγ and recombinant IFNγ treatments.** For anti-IFNγ treatment, 100 μg anti-IFNγ (Clone R4-6A2, Oregon Health and Science University Monoclonal Antibody Core) or IgG control (Sigma-Aldrich) was injected intraperitoneally every 3 days. For recombinant IFNγ treatment, 250 ng carrier-free recombinant mouse IFNγ (BioLegend) was injected intraperitoneally every other day.

**Glucose tolerance testing.** Mice were fasted for 6 h during the light phase with free access to water. A concentration of $2\,mg\,kg^{-1}$ glucose (Sigma-Aldrich) was injected intraperitoneally. Blood glucose was measured at 0 (immediately before glucose injection), 15, 30, 60 and 120 min with a Freestyle Lite glucometer (Abbot Diabetes Care).

**Food intake monitoring.** Mice were housed individually. Food weights were recorded every other day over a period of 1 week (four individual measurements), and average intake per day for each 2-day period was determined and averaged over the week measurement period for each individual.

**Bacterial DNA extraction and quantitative PCR.** For microbial DNA, frozen faecal pellets, caecal content and whole ileum with content were resuspended in 1.4 ml ASL buffer (Qiagen) and homogenized with 2.8 mm ceramic beads followed by 0.5mm glass beads using an OMNI Bead Ruptor (OMNI International). DNA was extracted from the entire resulting suspension using QiaAmp mini stool kit (Qiagen) according to manufacturer's protocol. DNA was quantified using Qubit broad range DNA assay (Life Technologies). A total of 10 ng of DNA was used for quantitative PCR (qPCR) reactions for *A. muciniphila* (AM1: 5′CAGCACGTGA AGGTGGGGAC′, AM2: 5′CCTTGCGGTTGGCTTCAGAT′)[52] and total bacteria (UniF340: 5′ACTCCTACGGGAGGCAGCAGT, UniR514: 5′ATTACCGCGG CTGCTGGC)[53]. qPCR was performed using Fast SYBR master mix (Applied Biosystems) and StepOne Plus Real Time PCR system and software (Applied Biosystems).

**RNA preparation and gene expression analysis.** RNA was extracted using an OMNI Bead Ruptor and 2.8 mm ceramic beads (OMNI International) in RLT buffer followed by Qiashredder and RNeasy kit using Qiacube (Qiagen) automated extraction according to manufacturer's specifications. Total RNA was quantified using Nanodrop (Thermo Scientific)[54]. Complementary DNA was prepared using iScript reverse transcription kit (Bio-Rad) and qPCR was performed using QuantiFast SYBR mix (Qiagen) and StepOne Plus Real Time PCR system and software (Applied Biosystems).

**16S rRNA gene sequencing and taxonomic analysis.** The V4 region of 16s rRNA gene was amplified using universal primers (515f and 806r)[55]. Individual samples were barcoded, pooled to construct the sequencing library, and then sequenced using an Illumina Miseq (Illumina, San Diego, CA) to generate pair-ended 250 nt reads. The raw forward-end fastq reads were quality-filtered, demultiplexed and analysed using 'quantitative insights into microbial ecology' (QIIME)[56]. For quality filtering, the default parameters of QIIME were maintained in which reads with a minimal Phred quality score of <20, containing ambiguous base calls and containing fewer than 187 nt (75% of 250 nt) of consecutive high-quality base calls, were discarded. Additionally, reads with three consecutive low-quality bases were truncated. The samples sequenced were demultiplexed using 12 bp barcodes, allowing 1.5 errors in the barcode. UCLUST (http://www.drive5.com/uclust)[57] was used to choose OTUs with a threshold of 97% sequence similarity against Green gene database (version gg_12_10)[58]. A representative set of sequences from each OTU were selected for taxonomic identification of each OTU by selecting the cluster seeds. The Greengenes OTUs

(version gg_12_10) reference sequences (97% similarity) were used for taxonomic assignment using BLAST[59] with $E\_value$ 0.001. Raw read counts of OTUs were normalized against total number of reads that passed quality filtration to generate relative abundance of OTUs. Differentially abundant OTUs were identified using univariate $t$-test in BRB array tools' 'class comparison between groups of arrays' module. BRB Array Tools was developed by the Biometric Research Branch of the National Cancer Institute under the direction of R. Simon (http://linus.nci.nih.gov/BRB-ArrayTools.html).

**RNA-seq.** RNA libraries were prepared with Wafergen Biosystems PrepX RNA-Seq Sample and Library Preparation Kits for the Apollo 324 NGS Library Prep System and sequenced using Illumina HiSeq. Forward read sequences with at least one ambiguous nucleotide were filtered out by prinseq[60]. Trimmomatic[61] was used to trim Illumina adaptor sequences (parameters: seed mismatches:1, palindrome clip threshold: 10 and simple clip threshold: 10), to remove leading and trailing low-quality bases (below quality 3), to scan the read with a 4-base wide sliding window, and to cut when the average quality per base drops below 20 and to drop reads that below 60 bases long. Reads were aligned to mouse genome and transcriptome (ENSEMBLE release-70) using Tophat[62] with default parameters. Number of reads per million for mouse genes were counted using HTSeq[63] and differential abundance of genes that are detected in at least three samples were analysed using edgeR[64].

**Network analysis and causal inference.** The gene–gene network was reconstructed using following steps. First, for each pair of genes, four Pearson correlation coefficients and $P$ values are calculated using expression levels of each of the four groups of mouse samples on different genetic backgrounds (C57BL/6($n = 8$), Swiss-Webster($n = 7$), B10A($n = 9$) and BALB/c($n = 10$))[15]. Second, Fisher combined $P$ value is calculated from four $P$ values using Fisher's combined probability test. FDR value is calculated from the combined $P$ value, that is, we calculate a test statistic as

$$\chi^2_{2k} \sim -2\sum_{i=1}^{k} \ln(P_i),  \qquad (1)$$

where $k =$ number of groups (4 in this example) and $P_i$ is the $P$ value of a single group (one mouse strain in this example). A $P$ value (the combined $P$ value) for $\chi^2$ is calculated under the fact that it follows a chi-squared distribution with $2k$ degrees-of-freedom. Third, the network is generated by selection and inclusion of gene–gene pairs as has been previously described[40]. Briefly, criteria for inclusion of gene–gene pairs are the following: Individual $P$ value of correlation within each group is $<0.3$; combined fisher $P$ value of all groups $<0.01$; the sign of correlation coefficients in four mouse strain groups should be consistent (all positive correlation or all negative correlation) and should be consistent with fold change relationship between the two genes (see 'Methods' section in ref. 23).

To prioritize genes that potentially mediate the effect of IFNγ on *A. muciniphila*, we integrated four sources of ranking information: Absolute value of the average correlation coefficient of ileal gene expression with ileal *A. muciniphila* abundance across three groups of IFNγKO mice; average shortest path length in gene–gene network (Fig. 4b); absolute value of log10 transformed fold change between their expression level in IFNγKO versus wild-type control mice; directional matching of correlation across three IFNγKO mouse groups, that is: assign positive sign to the absolute average correlation coefficient if the correlation signs are the same between all three groups, otherwise, assign negative sign to the absolute average correlation coefficient. Ranks from all four sources (corresponding values from smallest to largest) are summed to generate the final ranking of the genes.

**Human data from the ADVENTO study.** Subjects and protocol. A group of participants of the study Analysis of Diet and Lifestyle for Cardiovascular Prevention in Seventh-Day Adventists (ADVENTO—http://www.estudoadvento.org) conducted at University of São Paulo, Brazil was included in this cross-sectional study. The first 300 participants aged 35–65 years old were evaluated according to the eligibility criteria. Those with body mass index $\geq 40\,\mathrm{kg\,m^{-2}}$, history of inflammatory bowel diseases or persistent diarrhoea (longer than 2 weeks) and use of antibiotics or probiotic or prebiotic supplements within the 2 months before the data collection were not included. Five individuals were excluded from the final sample due to incomplete data. The University of São Paulo institutional ethics committee approved the study and written consent was obtained. Individuals were examined at the Investigation Center of University Hospital. After overnight fasting, they underwent a 2-h 75-g oral glucose tolerance test and American Diabetes Association criteria were used to define categories of glucose tolerance[37,38].

Analytical measurements. Blood samples were immediately centrifuged and analysed. Plasma glucose was measured by the hexokinase method (ADVIA Chemistry; Siemens, Deerfield, IL, USA) and HBA1c by high-pressure liquid chromatography (Bio-Rad Laboratories, Hercules, CA, USA).

Gut microbiota. Faecal samples were maintained under refrigeration (6 °C) within a maximum of 24 h after collection, when the aliquots were stored at −80 °C until analysis. DNA was extracted using the Maxwell 16 DNA purification kit and the protocol carried out in the Maxwell 16 Instrument according to the

manufacturer's instructions (Promega, Madison, WI, USA). Library preparation and sequencing were performed as described above for mouse samples. For analysis, high-quality sequences from 16S rRNA gene were obtained after trimming using Trimmomatic[61]. The paired reads were merged using FLASH[65] to reconstruct the contiguous sequenced region. Merged reads were then submitted to QIIME OTU picking pipeline[56]. The first step is the closed reference OTU clustering, using the GreenGenes[66] version 13.5. This same database was used to assign taxonomic classification to each OTU.

**Human data from the common variable immunodeficiency cohort.** Subjects and protocol. Duodenal biopsy samples were collected from healthy volunteers ($n = 4$) and immunodeficient patients with ($n = 7$) or without gastrointestinal syndrome ($n = 7$) at the National Institutes of Health Clinical Center. All protocol and consent procedures were approved by the National Institute of Allergy and Infectious Diseases and Oregon State University Institutional Review Boards. The diagnosis of common variable immunodeficiency and the associated enteropathy were made following international guidelines[67,68].

RNA isolation and sequencing. Total RNA was isolated using AllPrep kit (Qiagen) and prepared for sequencing using ScriptSeq Complete Gold Kit (Illumina). Four samples were pooled per lane and sequenced using paired end 100 bp Illumina Hiseq2000. Processing of raw data and analyses were performed as for mouse RNAseq except that reads were aligned with human transcriptome. For 16S rRNA gene sequencing, complementary DNA was prepared from RNA with SuperScript VILO kit (ThermoFisher), then amplified and sequenced using the same protocol as for mouse samples.

Correlation analysis. For analysis of correlation between *A. muciniphila* abundance and gene expression, we employed Pearson correlation in each one of the three patient groups followed by Fisher's combined probability test corrected by false discovery rate. To estimate a chance of given number of genes been negatively or positively correlated with *A. muciniphila*, we employed the test for one proportion implemented in https://www.medcalc.org/.

**Statistical analysis.** Statistical tests used for each comparison are described within corresponding figure legends. For large-scale data (transcriptome, microbiome and network reconstruction), statistical tests are described in the corresponding experimental procedures.

**Data availability.** Raw and processed data files for RNA-seq experiments have been deposited in ArrayExpress under accession code E-MTAB-3633. 16S rRNA gene sequencing raw read data have been deposited in the European Nucleotide Archive under accession codes PRJEB9551.

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

## Acknowledgements

This research was supported by startup funds for AM and NS from Oregon State University (OSU), USA; NIH U01 AI109695 (AM) and R01 DK103761 (NS) and by the Intramural Research Program of the NIH, NIAID (MY, IJF, WS). We also thank the FAPESP (São Paulo Research Foundation) for financial support (12/12626-9 and 12/03880-9) of ADVENTO Study. We thank Oregon State University Center for Genome Research and Biocomputing (CGRB) for sequencing services and technical support; Dr Daniel Cawley and the Oregon Health and Science University Monoclonal Antibody Core for production of IFNγ antibodies; and the personnel of Oregon State University Laboratory Animal Resources Center.

## Author contributions

A.M. and N.S. conceived the original idea, designed and supervised the studies and analysis, and wrote the manuscript. R.L.G. conceived the original idea, designed the studies, performed experiments, analysed the data and wrote the manuscript. X.D. designed the studies, performed microbiome and host transcriptome analyses, performed

causal inference analysis and wrote the manuscript. S.V.-P. conducted and assisted with mouse experiments. E.P. prepared libraries and provided other technical assistance. R.A.Z. and A.E.S. prepared *A. muciniphila* and participated in writing of the manuscript. A.C.F.M., G.R.F., E.P.G., A.C.P. and S.R.G.F. recruited human subjects of ADVENTO study, conducted clinical and microbiome analysis and participated in writing the manuscript. M.Y., I.J.F., W.S. recruited patients, conducted clinical evaluations, performed duodenal biopsy collection. A.A.S., G.A.T. and A.S.G. performed analysis of Irgm1KO mice and participated in writing of the manuscript.

## Additional information

**Competing financial interests**: The authors declare no competing financial interests.

**How to cite this article**: Greer, R. L. *et al. Akkermansia muciniphila* mediates negative effects of IFNγ on glucose metabolism. *Nat. Commun.* **7**, 13329 doi: 10.1038/ncomms13329 (2016).

**Publisher's note**: 

