## [Peer Review File · Nature Communications]

Reviewers' comments:

Reviewer #1 (Remarks to the Author):

Greer, Dong and colleagues have found that the abundance of one specific bacteria (*Akkermansia muciniphila*) and IFN gamma (using both IFN γ KO or treatment). They found that IFN γ KO mice have a different microbiota than WT mice. They have shown that the change in *Akkermansia* is linked to IFN γ , and that IFN γ mice lacking the bacteria are glucose intolerant as compared to *Akkermansia* positive mice. By using gene and prediction they propose that *Irgm1* controls the IFN γ /*Akkermansia* couple. Finally, the abundance of *Akkermansia* was correlated with glucose metabolism markers in humans, as well as gene expression in duodenal biopsies. The overall hypothesis is of interest, but there are numerous interpretations made although key controls are missing, therefore exposing the final conclusions to misinterpretations.

In sup Fig2B and C the authors claim that the increased abundance of *A. muciniphila* is detected in both the ileum and stool of IFN γ KO mice and that the levels in the stool are representative of those from the ileum; however, there are at least 2Logs of difference in the IFN γ Ko mice (ileum versus stools)? Moreover, the correlation shown in sup fig2C is not really conclusive. Why there are fewer points than in sup2B? In this figure there are only around 10 points that are represented? Please show the correlation with all the mice shown in supFig2B.

When carefully reading the taxa in sup tables, there are several hits that can also be of interest, for example the Lachnospiraceae, Rikenellaceae, or *Ruminococcus gnavus*, which was even more abundant than *Akkermansia* OTU's (higher by 5499 fold than in WT) (see also below the comments regarding littermates). It is worth noting that almost all these taxa are associated with obesity/diabetes in other studies. On line 181 to 187 the authors argue that there were no other candidates than *Akkermansia*, but again this is restricted to the glycemia, what about other key markers of obesity and diabetes? Fat mass? Insulin levels? Which are clearly lacking in all the ITT groups. Dealing with glucose metabolism require at least adequate markers, is there any change in insulin levels? Before after the tolerance test? Thus the overall hypothesis of the authors is oriented to bacteria and glucose metabolism, but why? Why not including correlations with other markers than glucose tolerance and *Akkermansia* abundance.

Another major confounding factor, probably jeopardizing the story is that the original mice used were not littermates. By comparing mice coming from a different colony (even the same supplier) can by essence introduce a bias. The microbiota can be different, the metabolism, the metabolites, the immune system and its response. The breeding experiments to eliminate *Akkermansia* in the KO is a key example showing that indeed breeding matters. What if the WT were Akk positive? It is surprising to see that the WT used in these experiments are *Akkermansia* negative, and never really tested for IFN γ levels or effects of *Akkermansia*/IFN γ others.

Thus a key question arising is : would the study and the conclusion have been the same if the mice were real littermates?

Along this line, IFN γ KO are leaner, is there any chance that this also contribute to the overall phenotype and microbiota? Many recent papers are suggesting this relationship.

In Fig2D, IFN γ treatment decreased the abundance of Akkermansia but what was the impact on other bacteria? Does IFN γ induce a general inflammatory tone? Any changes in antimicrobial peptides production in the gut that may contribute to the phenotype?

In figure 3, the gain of function is only shown in IFN γ KO why not in WT? Is there any beneficial impact of colonizing WT negative with Akkermansia? In other words, is this beneficial effect only restricted to IFN γ KO mice. Why the WT mice are excluded from the setting?

Are the results equivalent if Akkermansia is autoclaved? Is the phenotype partially restored?

What are the levels of IFN γ in these Akkermansia negative WT mice? Is IFN γ treatment inducing glucose intolerance in WT mice? It is plausible that although having basal levels of IFN γ the WT mice behave differently. Is Akkermansia able to improve glycemia in WT mice? (yes in sup fig4), but why as shown in the present study, the design always excluded the WT negative mice from manipulation with Akkermansia and IFN γ . Thus the key controls are missing? This should be shown. Although this is partially answered in SupFig4, this is also the proof that IFN γ is not the unique factor contributing to the phenotype? In addition, IFN γ levels are not shown in this experiment. A key experiment would be to administer antibodies against IFN γ to see whether blocking IFN γ in the presence of Akkermansia abolished or not its effects, this should be done in the WT mice.

The authors have compared parameters from human duodenal biopsies and Akkermansia, however, the major concern is that this bacterium is living in the lower part of the gut and not in this area of the gut highly exposed to oxygen. It is likely that doing similar correlation with Lactobacilli or Bifidobacteria may also occur. Do the authors have any evidence that Akkermansia was present in the duodenal biopsy?

Based on the comment above, what was the rationale for the inclusion of the human cohort in the present manuscript? Is it because the classification of the genes in fig5D showing IFN γ and related genes? According to the legend, the coefficients of correlation are all negative? Is it correct? From -1 to -0.1? please clarify the legend and results. Are the correlations significant?

Why having selected them as such? IFN γ is shown on the top of the list but it does not really seem to be the one who is the better correlated parameters, except the fact that the colour box in the HV appears as grey? Grey is not shown in the colours on the upper scale legends (-1 to -0.1), mostly blue.

Why not classifying the genes by statistical significance? Or by coefficient themselves?

Why not discussing other key markers such as body weight, fat mass, insulin levels? It is known that correlation exist between the abundance of Akkermansia and glucose metabolism (as already shown by other cohorts). But, is the correlation still significant when the three patients with the major abundance of Akkermansia are removed?

There is no clear discussion or results concerning the cohort in term of patients treatments. Several studies, have shown that metformin treatment changes the gut microbiota in favour of different bacteria including Akkermansia (Nature dec 2015). When looking carefully the correlation plot, one would suggest to apply a best fit curve, there are subjects with a really low level of Hb1c but also very low level of Akkermansia.

Thus, based on all the comments, the human data are not enough convincing to support such interactions between human gene expression, metabolism and Akkermansia.

Reviewers' comments:

Reviewer #1 (Remarks to the Author):

Greer, Dong and colleagues have found that the abundance of one specific bacteria (Akkermansia muciniphila) and IFN gamma (using both IFN γ KO or treatment). They found that IFN γ KO mice have a different microbiota than WT mice. They have shown that the change in Akkermansia is linked to IFN γ , and that IFN γ mice lacking the bacteria are glucose intolerant as compared to Akkermansia positive mice. By using gene and prediction they propose that Irgm1 controls the IFN γ /Akkermansia couple. Finally, the abundance of Akkermansia was correlated with glucose metabolism markers in humans, as well as gene expression in duodenal biopsies. The overall hypothesis is of interest, but there are numerous interpretations made although key controls are missing, therefore exposing the final conclusions to misinterpretations.

In sup Fig2B and C the authors claim that the increased abundance of *A. muciniphila* is detected in both the ileum and stool of IFN γ KO mice and that the levels in the stool are representative of those from the ileum; however, there are at least 2Logs of difference in the IFN γ Ko mice (ileum versus stools)? Moreover, the correlation shown in sup fig2C is not really conclusive. Why there are fewer points than in sup2B? In this figure there are only around 10 points that are represented? Please show the correlation with all the mice shown in supFig2B.

While overall, there is higher abundance of *A. muciniphila* in all IFN γ KO mice compared to their wild type controls, there is a wide range of *A. muciniphila* levels between experimental groups of IFN γ KO mice. Therefore, we chose to show a single group of IFN γ KO mice for the correlation analysis so that the analysis would not be confounded by combining high and low range groups into the same analysis.

When carefully reading the taxa in sup tables, there are several hits that can also be of interest, for example the Lachnospiraceae, Rikenellaceae, or Ruminococcus gnavus, which was even more abundant than Akkermansia OTU's (higher by 5499 fold than in WT) (see also below the comments regarding littermates). It is worth noting that almost all these taxa are associated with obesity/diabetes in other studies. On line 181 to 187 the authors argue that there were no other candidates than Akkermansia, but again this is restricted to the glycemia, what about other key markers of obesity and diabetes? Fat mass? Insulin levels? Which are clearly lacking in all the ITT groups. Dealing with glucose metabolism require at least adequate markers, is there any change in insulin levels? Before after the tolerance test? Thus the overall hypothesis of the authors is oriented to bacteria and glucose metabolism, but why? Why not including correlations with other markers than glucose tolerance and Akkermansia abundance.

We feel that the reviewer missed the discussion of our careful and unbiased computational approach for selecting *A. muciniphila* as a top candidate for further study. Of course there are other groups that are differentially abundant between wild type and IFN γ KO mice. However, simply because specific taxa are more abundant than *A. muciniphila* and/or more significantly different between two groups is a not sufficient criterion in our approach to warrant further study. Additionally, many of the taxa mentioned specifically by the reviewer were not altered only in one of our two analyses (wild type versus IFN γ KO

but not anti-IFN γ versus IgG). In order to eliminate possible artifacts and strengthen the conclusions that specific taxa are truly regulated by IFN γ , we required that the OTU be consistently differentially abundant in both datasets. Finally, our causal inference approach allowed us to narrow down our candidates by selecting only those that match the specific criteria described in the paper. We incorporated correlation to glucose metabolism and expected direction of correlation considerations into our selection model, which revealed *A. muciniphila* as our top candidate from this unbiased analysis (see Supplemental Figure 3).

Regarding the inclusion of additional metabolic markers – there are, of course, many factors that could be examined in addition to glycemia. However, we chose to focus on glycemia as our readout of metabolic state in this study. Additionally, as our mice are lean and normoglycemic, we do not find significant interest in looking at markers typically associated with obese states. In lean IFN γ KO mice, improved glucose tolerance during GTT was the most prominent observed difference compared to wild type (Wong, *Endocrinology* 2011) so we focused on that as our readout for this study.

Another major confounding factor, probably jeopardizing the story is that the original mice used were not littermates. By comparing mice coming from a different colony (even the same supplier) can by essence introduce a bias. The microbiota can be different, the metabolism, the metabolites, the immune system and its response. The breeding experiments to eliminate Akkermansia in the KO is a key example showing that indeed breeding matters. What if the WT were Akk positive? It is surprising to see that the WT used in these experiments are Akkermansia negative, and never really tested for IFN γ levels or effects of Akkermansia/IFN γ others.

Thus a key question arising is : would the study and the conclusion have been the same if the mice were real littermates?

We believe that the author overlooked the fact that all but a single experiment in this study was performed on littermates and/or mice of the same genotype. The initial non-littermate experiment was an exploratory phase experiment utilized to generate a hypothesis. This hypothesis was then tested using many different approaches that were not confounded by the issue of littermates (IFN γ reconstitution, IFN γ neutralization, *A. muciniphila* colonization).

It is clear that littermate studies can be informative. However, since microbiota composition is heavily determined by maternal transfer, we would also argue that littermate studies can mask or eliminate true phenotypes as well. In fact, our data indicates that a real effect of IFN γ on *A. muciniphila* (as demonstrated by IFN γ reconstitution and depletion studies on littermate mice of identical genotypes) is not observed in littermate studies on IFN γ KO mice derived from heterozygous parents, likely due to alterations in initial microbiota seeding (specifically a lack of *A. muciniphila* transfer to all offspring in our study). Therefore, in some cases breeding strategies can obfuscate results instead of clarify them. This is why we instead utilized methods that did not entirely rely on genetic IFN γ deficiency to prove the relationship between IFN γ and *A. muciniphila*. These methods, such as IFN γ reconstitution and neutralization, allowed us to compare effects of IFN γ within individuals, with each mouse serving as its own control. We believe that this is even more powerful than littermate analysis.

Additionally, we are not clear as to why the reviewer is surprised by the fact that some groups of wild type mice in our studies were *A. muciniphila* negative. And, as is the case in IFN γ KO mice, there is also a range of *A. muciniphila* abundance in wild type cohorts (see Fig S2), although still at a much lower abundance overall than IFN γ KO, so they are not always completely negative, just have low levels. The mice used for the experiments in Fig 3 were at the limit of detection for the qPCR assay, and were thus labeled as negative.

We are not sure what is meant by “effects of Akkermansia/IFN γ others” in this context, but we did test the ability of introduction of *A. muciniphila* to improve glucose metabolism in wild type mice as well (Fig. S4).

Along this line, IFN γ KO are leaner, is there any chance that this also contribute to the overall phenotype and microbiota? Many recent papers are suggesting this relationship.

While it is true that IFN γ KO mice are leaner than wild type when fed high fat diet (O’Rourke, Metabolism 2010), this relationship has not been established in lean mice. In fact, a previous study, while total body weight was slightly lower in the group of IFN γ KO mice tested, a trend towards increased fat pad mass was observed in IFN γ KO mice fed normal chow (Wong, Endocrinology 2011). In this study, we are using lean mice fed a normal chow diet. Additionally, our examination of weight showed that IFN γ KO mice are not consistently lower in body weight and found no clear relationship between total weight and improved glucose metabolism (Fig. S1).

In Fig2D, IFN γ treatment decreased the abundance of Akkermansia but what was the impact on other bacteria? Does IFN γ induces a general inflammatory tone? Any changes in antimicrobial peptides production in the gut that may contribute to the phenotype?

It is likely that IFN γ treatment would affect many different microbial taxa. We performed sequencing of mice before and after rIFN γ injection (mice shown in Fig. 3F-I). Although we observed some trends towards changes in microbe abundance, none were significant (FDR<10%).

Regarding inflammatory tone following IFN γ injection - The levels of recombinant protein used were optimized to reconstitute IFN γ KO mice to near wild type levels, not to induce an inflammatory response over what would be observed in wild type mice. Therefore, we do expect a change in the overall immune signature by reversal of changes that occur due to the loss of IFN γ , but only to the degree of restoring wild type signal. Measurements of serum IFN γ levels have been added to Figure 2.

Regarding “antimicrobial peptides” as we have demonstrated the key mediator of IFN γ effect on *A. muciniphila* is *Irgm1* we went back to global gene expression data from *Irgm1*KO and control mice. No apparent alterations in expression of Reg3g and defensins was noted. These result cannot rule out post-transcriptional regulation of antimicrobial peptides or other antimicrobial molecules.

In figure 3, the gain of function is only shown in IFN γ KO why not in WT? is there any beneficial impact of colonizing WT negative with Akkermansia? In other words, is this beneficial effect only restricted to IFN γ KO mice. Why the WT mice are excluded from the setting?

We did indeed colonize wild type mice with *A. muciniphila*, it was just a separate experiment (Fig. S4). Wild type colonization was not included in the study in Fig 3 because the action of *A. muciniphila* in wild type mice was not a primary question for this study, as well as due to limited numbers of littermate mice available.

Are the results equivalent if Akkermansia is autoclaved? is the phenotype partially restored?

It has been previously shown that live *A. muciniphila* is required for its improvement of glucose metabolism (Everard, PNAS 2013). Given that our improved glucose tolerance persists over the course of weeks and that we can alter glucose metabolism by IFN γ injection into mice colonized with *A. muciniphila* up to 3 weeks after colonization (Fig 3), we strongly believe that live colonization is required for full effects.

What are the levels of IFN γ in these Akkermansia negative WT mice? Is IFN γ treatment inducing glucose intolerance in WT mice? It is plausible that although having basal levels of IFN γ the WT mice behave differently. Is Akkermansia able to improve glycemia in WT mice? (yes in sup fig4), but why as shown in the present study, the design always excluded the WT negative mice from manipulation with Akkermansia and IFN γ . Thus the key controls are missing? This should be shown. Although this is partially answered in SupFig4, this is also the proof that IFN γ is not the unique factor contributing to the phenotype? In addition, IFN γ levels are not shown in this experiment. A key experiment would be to administer antibodies against IFN γ to see whether blocking IFN γ in the presence of Akkermansia abolished or not its effects, this should be done in the WT mice.

We disagree with the reviewers that experiments such as injection of wild type mice with IFN γ are key controls for this study. We were asking the focused question "in IFN γ KO mice, is *A. muciniphila* a mechanism of the improved glucose metabolism?". The wild type mice serve as a stable control, allowing us to better assess and normalize the effects of manipulation of the IFN γ KO groups. Inducing an increased inflammatory state by injection of IFN γ into WT mice that already have sufficient IFN γ levels is not a relevant control for this type of study as we have designed our experiments.

We do not claim that IFN γ is the unique factor controlling *A. muciniphila* and glucose metabolism, only that *A. muciniphila* is a mediator of the IFN γ effect on glucose metabolism. Our studies show that if *A. muciniphila* colonization can be established, it is capable of affecting glucose metabolism, whether in wild type or IFN γ KO mice. Our data clearly show that IFN γ is one factor that affects the abundance of *A. muciniphila*.

We do not agree that administering anti-IFN γ to wild type mice will be informative. As our data shows, our wild type mice do not have *A. muciniphila* at high abundance naturally. Even if they did, or we colonized wild type mice with *A. muciniphila* as done in Fig. S4, we do not see how neutralization of IFN γ in this context would further any conclusions. We would anticipate that this would allow an increase in *A. muciniphila* levels, thus possibly further improving glucose metabolism in those mice. We do not understand which effects the review anticipates would be abolished by this experimental setup.

The authors have compared parameters from human duodenal biopsies and Akkermansia, however, the major concern is that this bacterium is living in the lower part of the gut and not in this area of the gut highly exposed to oxygen. It is likely that doing similar correlation with Lactobacilli or Bifidobacteria may also occur. Do the authors have any evidence that Akkermansia was present in the duodenal biopsy?

A. muciniphila genomic DNA was clearly present in the duodenal biopsies, as it was present in microbiome sequencing results from these samples; however, we are not able to provide any evidence of viable *A. muciniphila* in these patients due to lack of material and the limited nature of the samples we have access to. We believe the reviewer may have misunderstood the samples used for this experiment - the IFN γ gene expression and microbiome sequencing were performed from the same biopsy sample. We do not understand why the reviewer assumes that a similar correlation between Lactobacilli or Bifidobacteria with IFN γ would also occur. What evidence is there for this conclusion? And even if these taxa do also correlate to IFN γ as *A. muciniphila* does, that does not detract any weight from our conclusion that a relationship exists between gut IFN γ levels and *A. muciniphila* levels in the human small intestine.

Based on the comment above, what was the rationale for the inclusion of the human cohort in the present manuscript? Is it because the classification of the genes in fig5D showing IFN γ and related genes? According to the legend, the coefficients of correlation are all negative? Is it correct? From -1 to -0.1? please clarify the legend and results. Are the correlations significant?

This human cohort allows us to examine the relationship between IFN γ and *A. muciniphila* within the intestine. We believe that these patient samples are a highly valuable resource which allows us to perform a strong and novel analysis. Thank you for pointing out to incomplete description of result and statistical analysis and mistakenly count extra gene. Now we added missing part of description. All concordant correlation coefficients are negative for (with 59 out 69 genes FDR<10% and if including all 69 genes FDR<24%; See details in Table S8), as this is the expected direction of relationship between IFN γ -dependent genes and *A. muciniphila*. As stated in the text, no gene from this network showed a consistent positive correlation (unexpected direction) with *A. muciniphila*.

Why having selected them as such? IFN γ is shown on the top of the list but it does not really seem to be the one who is the better correlated parameters, except the fact that the colour box in the HV appears as grey? Grey is not shown in the colours on the upper scale legends (-1 to -0.1), mostly blue.

As stated in the text and legend, the genes selected for this analysis are those identified as our IFN γ -dependent signature from IFN γ KO mouse ileum (Fig. 4). IFN γ itself does correlate, but the correlation of those genes downstream is stronger, as would be expected, since they are more peripheral to the network and thus closer to being direct mediators of *A. muciniphila*. Grey indicates undetected; this was added to the legend for clarity. IFN γ is a lowly expressed gene, and thus in our health volunteer group, it was below a threshold for detection. This fact shows another strength of our method of correlation to the entire IFN γ -dependent gene signature. It provides stronger evidence for a correlation between IFN γ signaling and *A. muciniphila* than correlation with the lowly expressed IFN γ gene alone.

Why not classifying the genes by statistical significance? Or by coefficient themselves?

The correlation results (except for IFN γ) are ranked by combined probability in three groups (FDR) that clearly reflects strength of correlations. IFN γ is intentionally shown on the top of the graph. We want reader not to miss the IFN γ ($r \approx -0.3$, $p = 0.127$) in the results showing that its own expression while tends to correlate with *A. muciniphila* correlates less strongly than IFN γ -dependent genes.

Why not discussing other key markers such as body weight, fat mass, insulin levels? It is known that correlation exist between the abundance of Akkermansia and glucose metabolism (as already shown by other cohorts). But, is the correlation still significant when the three patients with the major abundance of Akkermansia are removed?

Yes, with the 3 highest *A. muciniphila* abundance samples are excluded, the significant correlation still exists. However, we do not see the logic in excluding samples from this analysis. There is no scientific or statistical reasoning behind excluding patients with high *A. muciniphila*. Additionally, as the reviewer states, there is already significant evidence associating *A. muciniphila* to glucose metabolism. This data simply adds an additional cohort of supporting data to these conclusions. The strength and novelty of our human data analysis is that of the intestinal biopsy data proving correlation between IFN γ and *A. muciniphila*.

There is no clear discussion or results concerning the cohort in term of patients treatments. Several studies, have shown that metformin treatment changes the gut microbiota in favour of different bacteria including Akkermansia (Nature dec 2015). When looking carefully the correlation plot, one would suggest to apply a best fit curve, there are subjects with a really low level of Hb1c but also very low level of Akkermansia.

The patient cohort used for this study was newly diagnosed and had no previous therapeutic interventions. This has been clarified in the section of the paper discussing these patients as well as in the related figure legend.

Thus, based on all the comments, the human data are not enough convincing to support such interactions between human gene expression, metabolism and Akkermansia.

We disagree with the reviewer on this point, and believe that this conclusion was due to some misunderstanding of the duodenal biopsy cohort analysis as we have outlined in the responses regarding this dataset above.

REVIEWERS' COMMENTS:

Reviewer #2 (Remarks to the Author):

Most of the points raised by the reviewer have been addressed, and the revised manuscript became clearer than the previous version.

I have two comments that may help further improve the paper.

1. Do *Irgm1* KO mice exhibit improved glucose tolerance, similarly to IFN γ KO mice? This is important because current results are not sufficient to demonstrate the contribution of IRGM1 to the IFN-g-mediated regulation of *Akkermansia* abundance and glucose metabolism.
2. The phrases 'IFN-g-dependent microbes' and 'IFN-g-dependent mechanisms' used in some places in the text are misleading, because '-dependent' sounds positive effect by IFN. Those should be rephrased such as 'IFN-g-regulated'.

Point-by-point response to any issues raised by referees

Reviewer #2 (Remarks to the Author):

1. Do Irgm1 KO mice exhibit improved glucose tolerance, similarly to IFN γ KO mice? This is important because current results are not sufficient to demonstrate the contribution of IRGM1 to the IFN-g-mediated regulation of Akkermansia abundance and glucose metabolism.

-Regarding glucose tolerance in Irgm1KO mice, no such data are available in literature and we were not able to test these mice. This warrants future investigation.

2. The phrases 'IFN-g-dependent microbes' and 'IFN-g-dependent mechanisms' used in some places in the text are misleading, because '-dependent' sounds positive effect by IFN. Those should be rephrased such as 'IFN-g-regulated'.

-We agree that 'IFN γ -dependent' might sound misleading, especially related to *A. muciniphila* which is negatively regulated by IFN γ . Thus we changed to the suggested wording.

3. Change in response to one of the previous reviewer's question.

Question by previous Reviewer #1: "There is no clear discussion or results concerning the cohort in term of patients treatments. Several studies, have shown that metformin treatment changes the gut microbiota in favour of different bacteria including Akkermansia (Nature dec 2015). When looking carefully the correlation plot, one would suggest to apply a best fit curve, there are subjects with a really low level of Hb1c but also very low level of Akkermansia."

Our previous answer: "The patient cohort used for this study was newly diagnosed and had no previous therapeutic interventions. This has been clarified in the section of the paper discussing these patients as well as in the related figure legend."

Corrected answer: While rechecking all tables, we found that it was an incorrect statement in our previous answer. It appears that some patients were taking metformin (6 out of 11) at the time of stool collection. However, there is no difference in *A. muciniphila* abundance or metabolic parameters between subjects treated or not treated with metformin (Supplementary Figure 6). Therefore, the fact that some patients received metformin does not affect the overall results or conclusions of this section. In addition, what we see in small number of patients is actually in agreement with the large human study (mentioned by reviewer) that actually didn't find a consistent association between Akkermansia levels and metformin in humans (Forslund et al, Nature 2015) in contrast to mouse studies. We now added this result to the text (p. 12; Supplementary Figure 6).